# The winner takes it all—Competitiveness of single nodes in globalized supply networks

Chengyuan Han[1,2]*, Dirk Witthaut[1,2], Marc Timme[3], Malte Schröder[3]

**1** Forschungszentrum Jülich, Institute for Energy and Climate Research - Systems Analysis and Technology Evaluation (IEK-STE), Jülich, Germany, **2** Institute for Theoretical Physics, University of Cologne, Köln, Germany, **3** Center for Advancing Electronics Dresden (cfaed) and Institute for Theoretical Physics, TU Dresden, Dresden, Germany

* ch.han@fz-juelich.de

## Abstract

Quantifying the importance and power of individual nodes depending on their position in socio-economic networks constitutes a problem across a variety of applications. Examples include the reach of individuals in (online) social networks, the importance of individual banks or loans in financial networks, the relevance of individual companies in supply networks, and the role of traffic hubs in transport networks. Which features characterize the importance of a node in a trade network during the emergence of a globalized, connected market? Here we analyze a model that maps the evolution of global connectivity in a supply network to a percolation problem. In particular, we focus on the influence of topological features of the node within the underlying transport network. Our results reveal that an advantageous position with respect to different length scales determines the competitiveness of a node at different stages of the percolation process and depending on the speed of the cluster growth.

## Introduction

Global connectivity is central to our social, economic and technological development [1–4]. The growth of a global transportation network has dramatically changed world economy and led to increased efficiency and more centralized production [5]. But this global connectivity also bears new, systemic risks—highlighted in particular in the financial sector [6, 7].

Economies of scale are a major driving force in the formation of many of these socio-economic networks. Generally, a well developed economic agent with high connectivity is more attractive or competitive compared to smaller, less developed agents. The larger agents thus naturally attract even more connections [8–10]. In social network theory, this principle is commonly referred to as preferential attachment, driving the formation of scale-free networks [11]. In economic theory, economies of scale have been identified as a key mechanism leading to the emergence of trade networks and globalization [5, 12]. More recently, we have seen the emergence of quasi-monopolies in digital platform economies where economies of scale are particularly strong [13–15]. In this case the winner takes it all. But who wins and how?

German Federal Ministry of Education and Research (BMBF) (https://www.bmbf.de) Grants no. 03SF0472A-F, 03EK3055, and 03EF3055F. M. T. was supported by German Science Foundation (DFG) (https://www.dfg.de/) Center for Advancing Electronics Dresden (cfaed). The funders had no role in study design, data collection and analysis, decision to publish, or preparation of the manuscript.

**Competing interests:** The authors have declared that no competing interests exist.

Understanding which node in a network is the most competitive one and how it 'wins' over the competition as the network evolves toward global connectivity is still largely an open question. In particular, a systematic study of network formation in a heterogeneous geographic environment is a demanding task. Percolation models describing network growth typically involve random processes [16–18], while optimization models of the network structure typically start from a single global objective function [19–23]. However, neither model class fully describes socio-economic networks, whose formation is determined by the individual decisions (optimization, non-random) of interacting agents (multiple different objective functions). Economic equilibrium models and game-theoretic models capture these interactions and the individual decision but quickly become intractable as the number of agents increases [3, 24–28].

In this article, we study a simplified supply network model that explicitly includes nonlinear nonconvex economies of scale and transportation costs while simultaneously enabling a semi-analytical treatment by mapping the evolution of the network to a percolation problem [29]. In the model, agents try to satisfy a given demand at minimum costs, either through domestic production or via imports. Economies of scale favor the centralization of production and the emergence of trade. On the other hand, non-zero transportation costs favor distributed production. Simulating the evolution of the emerging trade network in this model allows us to systematically study how the transition to a globally connected supply network takes place, how the transportation network affects this transition, and last but not least which geographic factors provide an advantage for the competitiveness of the economic agents. In particular, we demonstrate that the way to be successful in the globalization process is to be in an advantageous position on the correct length scale. We show that the length scale characterizing the competitiveness of a node changes depending on the stage of the percolation process and the speed of the cluster growth.

## Methods

### Economic percolation model

We analyze the influence of topological features on the importance of nodes in a network formation model recently introduced by Schröder et al. [29]. The model describes the formation of global connectivity in networks inspired by the evolution of trade interactions in a fundamental network supply problem [5, 12]. The idea is as follows: Each node (or economic agent) $i \in \{1, 2, \ldots, N\}$ in the network has a fixed demand $D$ (identical for all nodes). A node $i$ can either fill this demand by domestic production or by making purchases from other nodes it is connected to via the underlying transport network. Filling this demand always incurs costs for node $i$: (I) production costs $K_{ki}^{\mathrm{P}}$ for production at node $k$, even for domestic production where $k = i$, and (II) transport costs $K_{ki}^{\mathrm{T}}$ for transport from node $k$ to node $i$ if node $i$ makes purchases from other nodes ($k \neq i$). This general setup is illustrated in Fig 1.

The production costs of goods manufactured at node $k$ and consumed at node $i$ are given by

$$K_{ki}^{\mathrm{P}} = p_k(S_k) \times S_{ki}, \tag{1}$$

where $S_{ki}$ denotes the amount of goods produced at node $k$ and consumed at node $i$. The costs per unit $p_k$ are *decreasing* with the total production $S_k = \sum_{i=1}^{N} S_{ki}$ due to *economies of scale* at node $k$. This means production becomes more efficient for larger quantities. Throughout this article we assume a linear relation

$$p_k(S_k) = b_k - a S_k \tag{2}$$

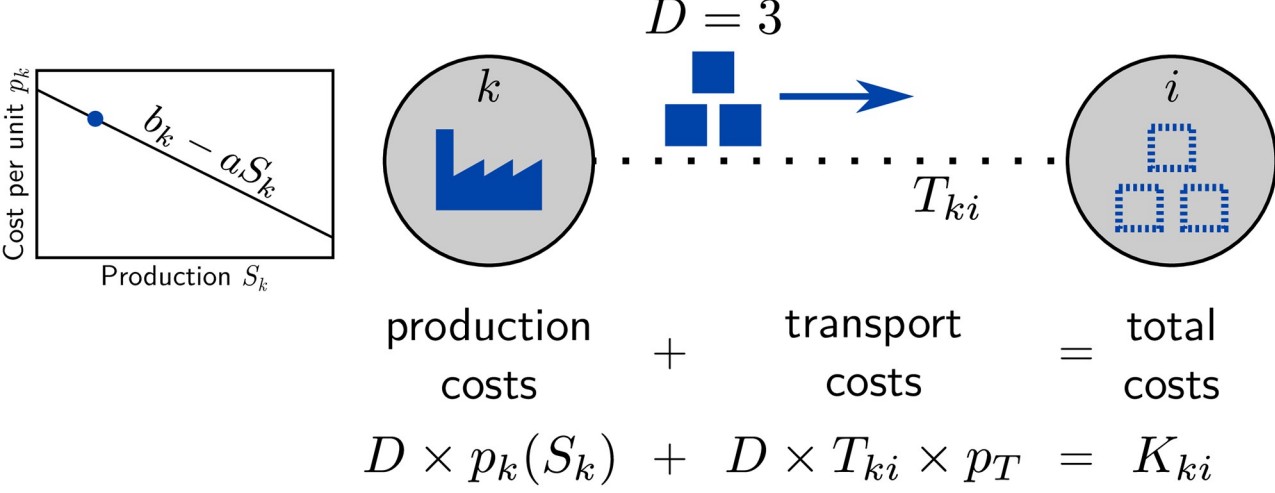

**Fig 1. Network supply problem.** Each node $i$ chooses a supplier $k$ to satisfy its demand $D$ at minimal cost $K_i = \min_k K_{ki}$. These costs include: (I) production costs at node $k$, where the costs per unit depend on the total amount of production $S_k$ at that node (left panel), and (II) transport costs that depend on the distance $T_{ki}$ between the nodes $k$ and $i$ in the underlying transport network (dashed line). All nodes in the network (including $k$) simultaneously solve their individual optimization problem.

for the sake of simplicity, where the parameter $a \geq 0$ directly quantifies the effective strength of the economies of scale and $b_k$ is a constant offset different for each node, describing inherent production cost advantages.

The transport costs

$$K_{ki}^{\mathrm{T}} = p^{\mathrm{T}} T_{ki} S_{ki} \qquad (3)$$

are proportional to the amount of purchased goods $S_{ki}$ and the distance $T_{ki}$ between the nodes in the underlying transport network. The proportionality factor $p^{\mathrm{T}}$ controls the importance of transport costs relative to production costs. In real-world settings, it typically decreases over time due to technological advancements in the transport sector and serves as the main control parameter for the network formation model. Together, the total costs for node $i$ read

$$K_i = \sum_{k=1}^{N} K_{ki} = \sum_{k=1}^{N} K_{ki}^{\mathrm{P}} + K_{ki}^{\mathrm{T}} \qquad (4)$$

as illustrated in Fig 1. This cost structure captures the fundamental incentives for the agents in this supply network percolation process.

Each node $i$ chooses its purchases $S_{ki}$ in order to minimize its costs under the constraint that it exactly satisfies its demand, $\Sigma_k S_{ki} = D$. In general, this leads to $N$ interacting nonlinear and nonconvex optimization problems as the production costs depend on the purchases of all (other) nodes. Nevertheless, a resulting Nash equilibrium, where no node can further decrease its costs by changing its supplier, can be computed efficiently as shown in [29]: Each node $i$ chooses only a single supplier $k$ (either itself or one other node in the network) that can be found efficiently with an adapted breadth-first-search due to the mapping to a local percolation problem. While multiple Nash equilibria exists for each value of $p^T$, this mapping uniquely defines the sequence of Nash equilibria describing the states of the supply network during the slow decrease of $p^T$ depending on the parameters and initial conditions.

We study the evolution of the supply network starting from the limit of infinite transport costs, $p^{\mathrm{T}} = \infty$, such that all nodes purchase locally and no trade takes place. As the

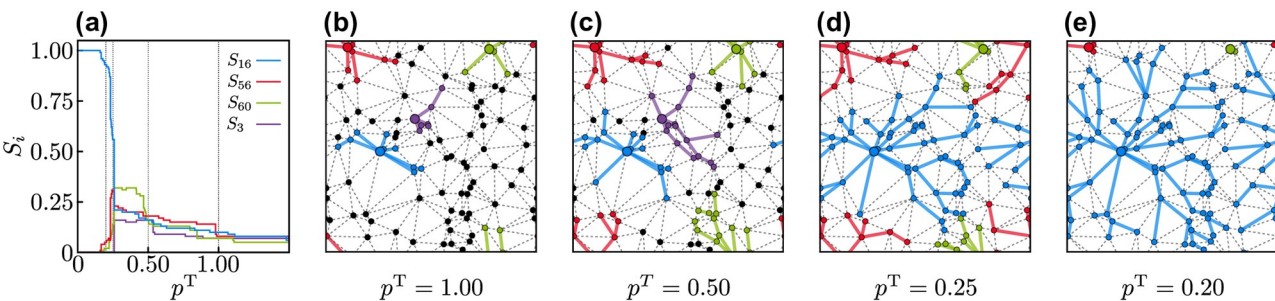

**Fig 2. Cluster growth in the percolation model.** (a) Evolution of the size $S_i$ of four clusters measured by the production $S_i$ of the clusters supplier $i$ (the number of nodes relative to the size of the whole network). Every node in the network optimizes its costs to satisfy its demand as described in the main text. As the importance of transport costs $p^T$ decreases, nodes make external purchases and clusters (common markets) emerge where production is centralized at a single node $k$. As $p^T \to 0$, only a single, global cluster with a central supplier $k^* = 16$ and $S_{16} = 1$ remains (blue line). (b-e) Snapshots of the network for different values of $p^T$. The four clusters with centralized production shown in panel (a) are illustrated in their respective colors and the central supplier node is highlighted. Black nodes do not belong to any of these four clusters. Solid colored lines indicate active links in the transport network, dashed lines indicate potential transport links that are not used by the four large markets. Parameters are $D = 1/N$, $b \in [0, 1]$ distributed uniformly at random and $a = 10^{-3}$. The planar network is created as the Delaunay triangulation from $N = 100$ points distributed uniformly at random in the unit square (see Methods for more details).

importance of transport costs decreases, some nodes start to make non-local purchases such that the production $S_k$ of other nodes increases. Eventually, large common markets (clusters) emerge in the network of trades $S_{ki}$, each with a single supplier node $k$. In the end, when transport costs disappear, $p^T = 0$, only one giant cluster remains with a single supplier $k^*$ with globally centralized production $S_{k^*} = ND$. This evolution is illustrated in Fig 2 for a small planar network.

In this article we study two main aspects of the formation of this trade network: First, how does centralization occur? That is, how does the transition from local production at large $p^T$ to centralized production at low $p^T$ take place? Second, we analyze which node $k^*$ becomes the final supplier (the center of the globally connected cluster) as production becomes fully centralized for $p^T \to 0$.

## Analysis of network structure

The economic percolation model includes heterogeneous geographical conditions explicitly. The matrix $T_{ki}$ encodes the distances of all pairs of nodes $(k, i)$ which depends on their geographic location and the structure of the underlying transportation network. Hence, the model allows to systematically study the influence of geographical or topological properties on the formation of connectivity and trade and the centralization of production. Are there any geographical or topological features that determine which node becomes the final supplier and which does not?

To study the impact of the transport network topology, we consider four different random network ensembles. We start from an ensemble of geographically embedded networks obtained by distributing $N = 1000$ nodes uniformly at random on the unit square. Edges are constructed by a Delaunay triangulation with periodic boundary conditions. Each of the resulting $M = 3000$ links is undirected and assigned a distance equal to the Euclidean distance between the connected nodes. The distance $T_{ij}$ of two arbitrary nodes $i$, $j$ in the network is finally obtained as the geodesic or shortest path distance in the network.

The other random network ensembles are obtained from the initial ensemble by a reshuffling of the edges. This procedure keeps the number of connections and the distribution of the individual edge lengths identical and thus leaves the networks comparable to each other. We apply three different reshuffling procedures creating randomizations with different properties:

First, we keep the structure of the network the same but choose a random permutation of the distances (random weights). This breaks correlations between the link distances and the node position. Second, we uniformly randomly rewire all links to different nodes under the constraint that the resulting network is connected. The network then has a topology corresponding to a Poisson random network [2]. Comparison of this randomization to the original network allows us to understand the impact of regular versus random network topologies. Third, we create a Barabasi-Albert scale-free network with the same number of links and the same distances for the links [11]. We thus create four different ensembles with identical average degree and edge lengths, but vastly different global structures. For instance, the degree distribution changes from narrow for the geometric and Poisson random networks to heavy-tailed for scale-free networks.

## Model parameters

In addition to the structure of the transportation network, several model parameters determine the evolution of the trade network. First, we note that the system evolution is invariant with respect to a rescaling of the costs. In particular, we can set $D = 1/N$ by choosing an appropriate unit system. A rescaling of the distances can be absorbed into the main control parameter $p^T$ describing the transport cost per unit. It characterizes the *relative* importance of transportation costs with respect to production costs.

Two parameters $a$ and $b$ characterize the production costs via the costs per unit $p(S_k) = b_k - aS_k$ [Eq (2)]. Since only the relative ordering of the costs are relevant to compare different suppliers (in the form of $K_{ki} < K_{ji}$), we scale the costs such that all $b_i \in [0, 1]$ with $\min_i b_i = 0$ and $\max_i b_i = 1$. In particular, we choose the $b_i$ uniformly at random from the interval [0, 1]. The second parameter $a$ characterizes the economies of scale and has a strong impact on the model behavior. We perform simulations for vastly different values $a \in \{10^{-5}, 10^{-4}, \ldots 10^1\}$ to cover all different regimes. To put this into context, note that total centralization of production leads to a decrease of production costs by exactly $NDa = a$ for $D = 1/N$. Economies of scale are negligible if $a$ is much smaller than typical differences of the cost parameter $b_i$, i.e., for $a \ll 1/N = 10^{-3}$. Economies of scale are dominant if $a$ is of the order of the largest difference of the $b_i$, i.e. for $a \approx 1$. The range $a \in \{10^{-5}, 10^{-4}, \ldots 10^1\}$ covers both regimes.

In summary, we perform simulations for four different transportation network ensembles and several values of $a$. For each case we consider 1000 different random realizations of the transportation network with 10 different permutations of the $b_i$ each, resulting in 10.000 measurements per ensemble and value of $a$. For each realization, we start the simulation in the limit of large transport costs, $p^T = \infty$, without any trade interactions. We gradually lower $p^T$ and record the emergence of a trade network, i.e., the emergence of connected components of the network defined by the purchases $S_{ki}$, as well as the final supplier for $p^T = 0$.

## Results

### How does global connectivity emerge?

To understand the emergence of a globally connected network we record the size of the largest clusters as the transport costs decrease from $p^T = \infty$ (no trade) to $p^T = 0$ (single, globally connected cluster). A trade network between nodes emerges as transportation costs decrease. An example of the centralization of production is shown in Fig 2 for a small geographically embedded random network. For $p^T = 1.0$, several nodes have already decided to purchase their goods from other neighboring nodes and multiple clusters have formed where production is

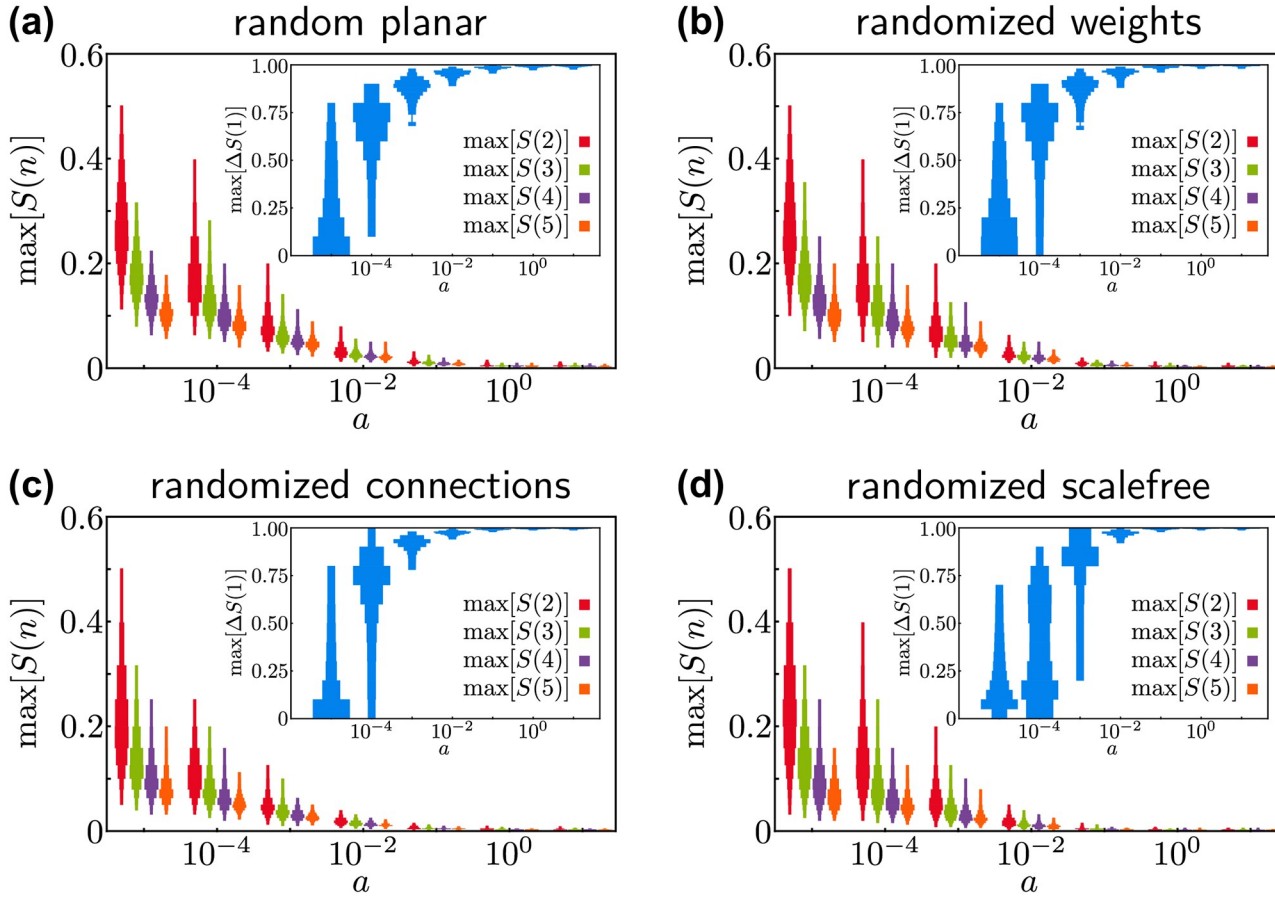

**Fig 3. Multiple clusters or sudden growth?** Distribution of the maximum size $\max[S(n)]$ of the $n$-th largest cluster and largest change $\max[\Delta S(1)]$ in the size of the largest cluster (insets) during the emergence of global connectivity for (a) the random planar network, (b) the network with randomized weights, (c) the network with uniformly randomized links and (d) the network with scale-free randomized links. For small $a$, multiple large clusters appear and merge slowly in all networks. For large $a$, a globally connected cluster suddenly forms from the individual nodes in a single large cascade before any other cluster had the chance to grow significantly. Depending on the value of the parameter $a$, nodes have to be competitive at different length scales to become the final supplier. The maximal size of the second largest cluster $\max[S(2)]$(red) can serve as a proxy for this length scale.

centralized to a single node. The clusters grow when $p^{\mathrm{T}}$ decreases to $p^{\mathrm{T}} = 0.5$ as further nodes decide to purchase non-locally. Finally, many nodes again change their supplier, joining one large, global cluster with strong economies of scale instead of the smaller local clusters. In the end, as $p^{\mathrm{T}} \to 0$, production is fully centralized at a single node. The size of the four largest clusters is shown in Fig 2(a) as a function of the transportation cost parameter $p^{\mathrm{T}}$.

Inspecting this evolution, we are directly led to the question how the transition to global connectivity takes place under different circumstances. Is it very sudden with a single large change in the size of the largest cluster or is the transition slow and the largest cluster grows gradually as $p^{\mathrm{T}}$ decreases? Does a single node expand its cluster or do multiple large clusters grow and only later merge to one global cluster? To answer these questions, we measure the largest gap $\max[\Delta S(1)]$ in the size (total production) of the largest cluster [30] as well as the maximum size of the second largest cluster $\max[S(2)]$, the third largest cluster $\max[S(3)]$ and so on over the course of the evolution from infinite to zero transport costs (see Fig 3). The maximal size $\max[S(2)]$ of the second largest cluster in particular measures how much clusters grow before global centralization occurs. If it is small, only a single large cluster emerges and local competitiveness is relevant to gain an early advantage. If it is large, at least two large

clusters expand side by side before one of them becomes globally dominant and production is completely centralized. Here, the central nodes of the clusters have to compete against each other on a larger length scale. The maximal size max[$S(2)$] of the second largest cluster serves as a proxy for this length scale.

If economies of scale are weak (small values of $a$), multiple large clusters coexist before they finally merge. As $a$ becomes larger, the maximum size of all clusters except the largest one decreases. Finally, for strong economies of scale $a$, only a single cluster grows. Correspondingly, the transition to global connectivity becomes more and more abrupt with increasing $a$, measured by the growth of the gap max[$\Delta S(1)$]. We thus obtain the following picture: For weak economies of scale, several clusters grow and finally merge in a gradual process. For strong economies of scale, only local clusters exist until a globally connected cluster emerges in abruptly. After this sudden transition, exactly one globally connected cluster remains.

We observe rather little differences between the four network ensembles under consideration. The transition from gradual to abrupt emergence of global connectivity is qualitatively the same in all networks and also the transition point is remarkably similar. While the transition is gradual (no large gaps) for $a = 10^{-5}$, it is sudden for $a = 10^{-3}$ for all networks. Slight differences are observed only for $a = 10^{-4}$. While the maximum gap is larger than 0.1 for all realization of the random planar network, the transition is still gradual with smaller changes of the largest cluster for most realizations of a scale-free network.

This is rather surprising, as scale free networks are characterized by the existence of hubs, a few nodes with very high degree. At first glance, one might expect that these hubs can exploit economies of scale most easily, making the transition abrupt already for small $a$. Our results show that this simple reasoning fails. The impact of economies of scale on the transition and on the competitiveness of nodes is more subtle. In fact, different hubs have to compete when the economies of scale are not dominant (small $a$). Thus, while hubs allow for the easier formation of local clusters, these hubs then have to compete on a larger length scale (measured by the maximum size of the second largest cluster), where the local properties of the central supplier, such as the high degree of the hubs, are less important. Overall, this competition slows down the centralization of production in scale-free networks. This idea is similar to the mechanism preventing or delaying the merger of large clusters in models resulting in explosive and discontinuous percolation transitions [18, 31, 32].

## Who becomes the central supplier?

Understanding *how* global connectivity emerges, we now address the question *who* wins the competition in this model. That is, which node $i$ becomes the central supplier of the network for $p^{\mathrm{T}} \to 0$? Are there any geographic features that determine a node's competitiveness?

To characterize the geographical location of a node in a network, we consider several different centrality measures that measure different aspects of a node's position in the network:

(i). cost centrality $1/b_i$

(ii). local closeness centrality $1/min_j T_{ij}$

(iii). global closeness centrality $1/\sum_j T_{ij}$ [33, 34]

(iv). degree centrality [34]

(v). betweenness centrality [34, 35].

These quantities measure the advantage of the nodes in terms of (i) global production costs, (ii) small transport costs to a local trade partner, (iii) small transport costs to the whole network, (iv) immediate access to different trade partners and (v) position of the node along many trade routes.

We generally expect that all these properties are beneficial for the nodes. For example, a high cost centrality implies that production is cheap—at least until production costs decrease significantly due to economies of scale. The node with the highest cost centrality would be the socially optimal supplier when $p^{\mathrm{T}} = 0$ and minimize the total costs across all nodes. Similarly, a high global closeness centrality implies that transportation is cheap on average, making the node an attractive global supplier when transport costs are not zero. The remaining three centrality measures also point to a favorable position in the network, but their implication is less clear. High degree and local closeness point to an attractive local environment, while high betweenness centrality is a typical measure of importance in social networks and means that many shortest transportation routes cross the respective node.

To understand which of these properties most strongly influences the competitiveness of a node, we rank all nodes according to their centralities and evaluate if the final suppliers typically have a high or low ranking. We record the final supplier and its centrality ranking $x$ for each random realization of the percolation process. The resulting distributions of the ranks of the final supplier are shown in Fig 4 for the four network ensembles under consideration. In addition, we fit a distribution $P(x) \sim \exp[-m(N - x)]$ to the observed centrality rankings to quantify the importance of the respective centrality. A value of $m = 0$ indicates a flat distribution, i.e., no influence of the centrality rank $x$ on the chance to become the final supplier. The higher the value of $|m|$, the stronger the correlation, and the more meaningful the respective centrality to predict which node becomes the central supplier.

The first, expected observation is the influence of the cost centrality $1/b_i$ of a node $i$. For weak economies of scale (small $a$) the production costs are dominated by the cost parameters $b_i$ and low production costs are decisive for the competitiveness of a node. For all network ensembles under consideration, cost centrality is the best indicator for competitiveness for small $a$, whereas its importance decreases for stronger economies of scale.

The second, more striking observation is the importance of the local closeness centrality. In the case of strong economies of scale $a = 1$, this centrality measure provides the best indicator for the competitiveness of a node. The histogram of the centrality ranking peaks strongly at top ranks. Local closeness is even more important than global closeness, although we evaluate the global competitiveness of the nodes. Again, this finding holds true for all four network ensembles.

A surprising correlation is found for the two remaining centrality measures, degree and betweenness, for the spatially embedded random network. Contrary to our expectation, the final supplier typically has a *low* degree and betweenness centrality for strong economies of scale $a$. This effect is lost or even reversed for the other network ensembles and can be attributed to a subtle geometric property of spatially embedded random networks. In this network class, local closeness centrality is anti-correlated with degree and betweenness centrality. As competitive nodes have a high local closeness, they are likely to have a low degree and betweenness centrality. This observation is particularly relevant since real-world transportation networks are typically spatially embedded, with the exception of digital, data exchange networks. Note that similar correlations exist for other network ensembles as well. For example, nodes with a high degree centrality in the reshuffled scale free networks typically also have high local closeness centrality, due to more opportunities for a short link.

Finally, a more subtle implication of the centrality measures is that, depending on the parameter $a$, the size or length scale of the relevant neighborhood changes. This length scale is

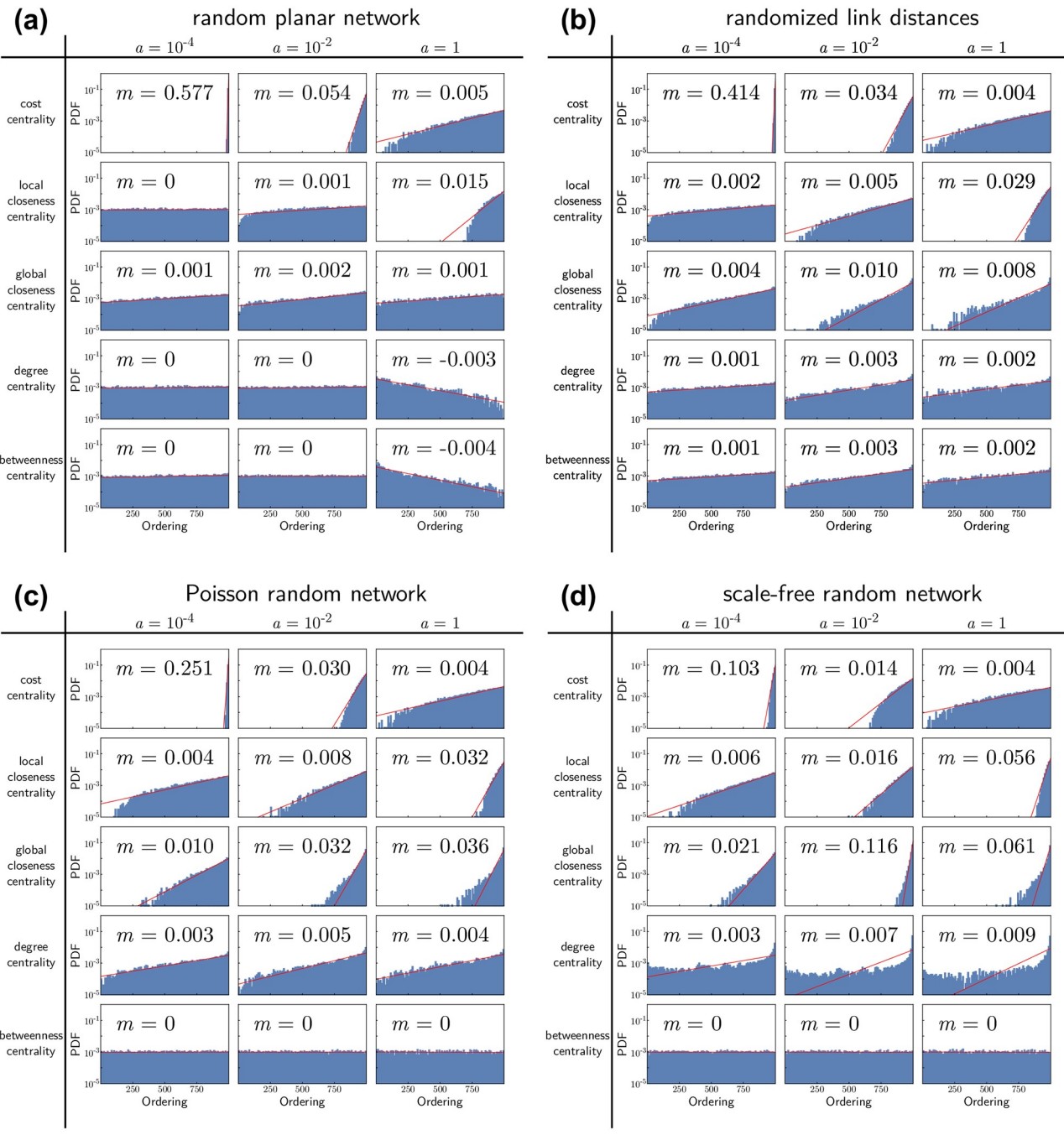

**Fig 4. How to become the central supplier?** Distribution of the ranking of the final supplier in various centrality measures (see main text) in (a) a random planar network, (b) the network with a random permutation of edge distances, (c) a Poisson random network with a random permutation of the edge distances, and (d) a scale-free network with a random permutation of the edge distances. All networks are constructed from a Delaunay triangulation of $N = 1000$ points uniformly randomly distributed in the unit square, resulting in $M = 3000$ links with distances equal to the Euclidean distance between the connected nodes (see Methods for details).

defined by the critical size the largest cluster must reach before it becomes the global supplier. The effect is illustrated in Fig 5. For small $a$, the number of customers does not significantly affect the costs and one new customer allows the supplier to attract customers only in a small additional range [Fig 5(a)]. Consequently, a node must attract a larger number of customers to

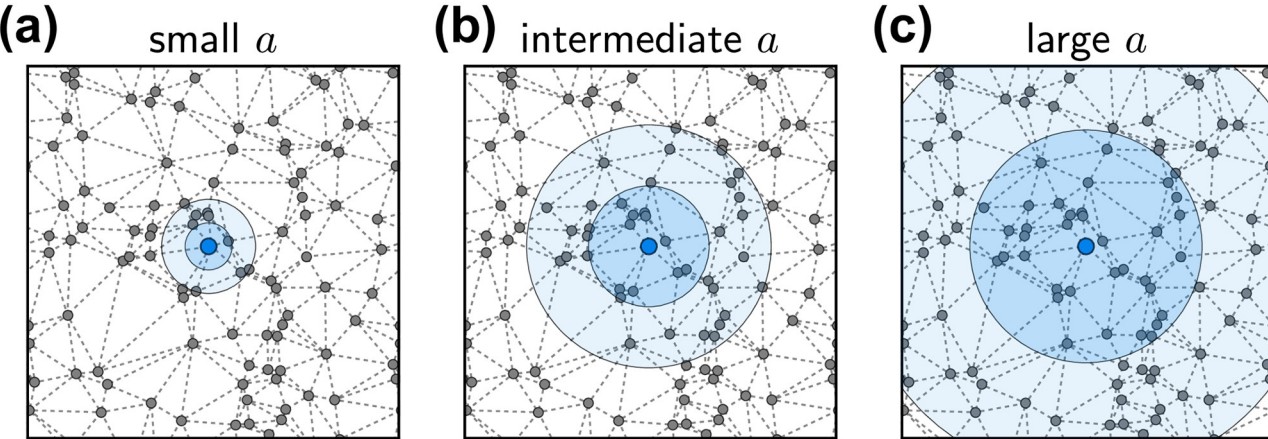

**Fig 5. Impact of a single customer.** Sketch of the effect of a single (new) customer for a node. With the new customer production increases and the production costs per unit decrease by $aD$ (economies of scale). This compensates larger transport costs for nodes further away from the supplier. Consequently, the supplier becomes competitive in a larger range and can potentially attract additional customers. The blue disks indicate the distance that is compensated by the decrease in production costs due to one customer (two customers). (a) For small $a$, the change in production cost is small and likely has no immediate effect [compare $a = 10^{-4}$ in Fig 4(a)]. The nodes have to compete at all length scales. (b) For intermediate $a$, a single customer may reduce the costs sufficiently to cause additional nodes to change their supplier. In this case, nodes have to compete at a local scale until they reach a size sufficiently large to take over the global cluster. (c) For large $a$, a single customer definitely reduces the costs sufficiently to cause a cascade of purchasing decisions and the first node to attract a customer takes over the whole cluster. Here, only the immediate neighborhood of a node decides about its success [compare $a = 1$ in Fig 4(a)].

become globally competitive and the critical size is (almost) equal to the total size of the network. In this regime, global centrality measures like the cost centrality are most relevant. For intermediate $a$, a single customer allows the supplier to attract nodes in a larger range [Fig 5(b)]. The critical length scale becomes smaller and we need to put more weight to the local structure. In this regime, the global closeness centrality and the degree centrality start to become better predictors, quantifying the centrality of a node in a local neighborhood. Finally, for very large $a$, the critical size of the largest cluster becomes 2 and one single customer induces a sufficiently large change in production costs for the supplier to become globally competitive immediately [Fig 5(c)]. The centrality of a node in its most local context then becomes the deciding factor. This is best measured by the distance to the nearest neighbor, the local closeness centrality $1/\min_j T_{ij}$.

Comparing results across the different network topologies, we find that the network topology becomes more important when the diameter is smaller, i.e., for Poisson and scale-free network structure. Since the total transport costs in these networks are smaller (proportional to the smaller diameter of these networks), the critical size to become the global supplier is also smaller. Thus, local length scales and the (local) network structure become important already for smaller values of $a$.

## Conclusion and discussion

Economies of scale are a decisive factor in the formation of socio-economic networks and the globalization and centralization of economic activities. Eventually, the winner takes it all. Here we have studied core aspects of the question who wins and how in a simplified model of supply network percolation.

The formation of socio-economic networks is a guiding research question across disciplines, including economics [4–6, 12], sociology [3, 27, 36] and statistical physics [2, 11]. Key mechanisms and global properties of network formation through economies of scale have been thoroughly analyzed [5, 11, 27], whereas the microscopic processes in large systems with

many heterogeneous actors are much harder to grasp. Most traditional models of network formation do not explicitly capture the behavior of individual actors [11, 17, 37]. Percolation models are based on random processes, while optimization models typically assume a common global objective function. In contrast, game theoretic models describing individual agents [21, 25, 26, 38] are often hard, if not impossible, to solve for large heterogeneous systems. In this article, we have analyzed a supply network model [29] that explicitly includes economies of scale and individual decisions, yet remains simple enough to allow for an efficient simulation of network formation and centralization in large heterogeneous environments. We exploit this fact to reveal the topological properties that determine the importance of a node for the emerging globally connected network.

The model yields the structure of a trade network given an underlying transportation network as a function of two main parameters: the strength of economies of scale $a$ and the transport costs per distance $p^{\mathrm{T}}$. As transport costs decrease, trade links are established and the production is centralized to fewer and fewer nodes. For weak economies of scale, this process is gradual. Nodes compete at all length scales and the merger of two large clusters is inhibited while transport costs are large, similar to mechanisms of explosive percolation [18, 31, 32]. The internal cost parameters are decisive for the competitiveness of a node. Only nodes with low productions costs $b_i$ have a chance to become the final supplier of the network once production is centralized completely. The geographic location of the nodes in the network, characterized by different centrality measures, plays only a minor role. In contrast, if economies of scale become dominant, this picture changes entirely: Production is centralized in a single, discontinous percolation transition once transportation costs decrease below a critical value. Only a single node attracts a significant number of customers and wins the competition almost instantly. Moreover, the transition becomes abrupt and as such hard to foresee. The chance of a node to become the central supplier is now mostly determined by the location of the node in the network. Interestingly, however, global centrality measures are not the best indicator for competitiveness. Instead, a local measure of the distance to the nearest neighbor, referred to as local closeness, is the best indicator for the success of a node. These results remain qualitatively unchanged for a broad range of cost functions describing economies of scale [29]. While modifications, for example stopping the process at non-zero transportation costs, change the quantitative evolution, the mechanistic insights into which length scales determine the importance of nodes during the emergence of (global) connectivity are generally applicable.

Loosely speaking, our findings are as follows: For weak economies of scale the internal properties of a node or economic agent are decisive. Competition occurs across all length scales in the network and basic efficiency provides the greatest advantage in all stages of the emergence of global connectivity. Only the (globally) most efficient nodes have a chance to take over the network. For strong economies of scale speed becomes the most important factor, rather than efficiency or global location. Competition occurs only locally to gain a first advantage and only the agent with the highest local closeness can rapidly attract the first external customers and then exploit economies of scale to grow its market, skipping over the competition in later stages of process. For the future it would be of eminent interest to study how other factors influencing economic globalization processes confirm or modify these findings and whether they can be confirmed in real world settings.

## Supporting information

**S1 Table. Information on the realization of network typologies (10 different realizations for each reshuffling method) indexed by *r*.** Legends can be found in te readme.txt file.
(ZIP)

**S2 Table. Simulation results.** Legends can be found in te `readme.txt` file.
(ZIP)

## Author Contributions

**Conceptualization:** Dirk Witthaut, Malte Schröder.

**Formal analysis:** Chengyuan Han, Malte Schröder.

**Funding acquisition:** Dirk Witthaut, Marc Timme.

**Investigation:** Chengyuan Han, Malte Schröder.

**Methodology:** Chengyuan Han, Dirk Witthaut, Malte Schröder.

**Project administration:** Dirk Witthaut, Malte Schröder.

**Resources:** Chengyuan Han, Dirk Witthaut, Marc Timme, Malte Schröder.

**Software:** Chengyuan Han, Malte Schröder.

**Supervision:** Dirk Witthaut, Malte Schröder.

**Validation:** Chengyuan Han, Malte Schröder.

**Visualization:** Malte Schröder.

**Writing – original draft:** Dirk Witthaut, Malte Schröder.

**Writing – review & editing:** Chengyuan Han, Dirk Witthaut, Marc Timme, Malte Schröder.

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
