## [Decision Letter · Decision Letter 0]

10 Aug 2019

PONE-D-19-16528

The winner takes it all - How to win network globalization

PLOS ONE

Dear Mr. Han,

Thank you for submitting your manuscript to PLOS ONE. After careful consideration, we feel that it has merit but does not fully meet PLOS ONE’s publication criteria as it currently stands. Therefore, we invite you to submit a revised version of the manuscript that addresses the points raised during the review process.

We would appreciate receiving your revised manuscript by Sep 24 2019 11:59PM. To enhance the reproducibility of your results, we recommend that if applicable you deposit your laboratory protocols in protocols.io, where a protocol can be assigned its own identifier (DOI) such that it can be cited independently in the future. For instructions see: http://journals.plos.org/plosone/s/submission-guidelines#loc-laboratory-protocols

We look forward to receiving your revised manuscript.

Kind regards,

Samir Suweis, Ph.D.

Academic Editor

PLOS ONE

Journal Requirements:

Additional Editor Comments (if provided):

Please answer carefully comments of reviewer 2 and also try to consider the comments of reviewer #1 for the title and introduction of the paper, avoiding over-claims

Reviewers' comments:

Reviewer's Responses to Questions

**Comments to the Author**

1. Is the manuscript technically sound, and do the data support the conclusions?

Reviewer #1: No

Reviewer #2: Yes

2. Has the statistical analysis been performed appropriately and rigorously? 

Reviewer #1: Yes

Reviewer #2: Yes

3. Have the authors made all data underlying the findings in their manuscript fully available?

Reviewer #1: Yes

Reviewer #2: Yes

4. Is the manuscript presented in an intelligible fashion and written in standard English?

Reviewer #1: Yes

Reviewer #2: Yes

5. Review Comments to the Author

Reviewer #1: A serious problem with this paper is that it has very little to do with trade & globalization, and is therefore misleading as far as it's title/advertising goes. The formulation of the problem is a very narrow, cost side approach, whereas "globalization," it's structure and who "wins" (the use of this term is questionable too), involve both demand and supply side considerations. More realistic additions to the model, such as the addition of variable costs on the production side (at the moment there are only fixed costs) and fixed costs on the transportation side (at the moment there are only variable costs) would invalidate the results of the model. In this sense, the model presented in the paper is a caricature of trade and globalization.

More specifically:

-The supply problem is divorced from the demand side. Why is this about globalization? The authors provide no interpretation or intuition of this.

-The production side of the economy is assumed to have a fixed cost, and the economies of scale come from this sole feature. There are no variable costs of production in the model. Such a cost structure is restrictive and applies to very few industries in the real world. There is no explanation for this omission.

-Transportation costs on the other hand are modeled as variable costs. Why? There could very well be fixed costs on the transportation side too. A small change such as this would invalidate the results.

The problem in the paper is more related to a cost-minimization engineering problem, that perhaps fits in logistics or operations research, but is a caricature of the economics of trade and globalization. It is a very narrow and misleading characterization of the problem.

Reviewer #2: In this work, the authors assess the importance of node in trade networks using a percolation-type model of economy of scale. They answer two questions: how a globalized market emerges and who wins the competition and takes the all in the globalization. They conclude two different scenarios depending on the strength of economies of scale. For weak economies of scale, internal properties (costs) of nodes is an important factor while for strong economies of scale, the efficiency is an important factor to find out the power of nodes.

Identification of the power of node in terms of the network structure is a central problem in complex network society and physics of complex systems. In this work, the authors successfully provide a method to identify and analyze the importance of nodes. This work is novel and all the results and discussion in the paper are well supported by their analysis. Therefore, I recommend the paper to be accepted in PloS ONE with a minor revision taking into account the following issues.

1. All the analysis in the paper is dependent on a resulting Nash equilibrium state in the model. Is this equilibrium state uniquely defined? If not, is the following analysis reliable?

2. The mechanism of merging processes with weak economies of scale is reminiscent the explosive percolation which have been popular in physics community recently. Can the authors add some discussion about a association between them?

3. The first sentence in the very last paragraph is not complete: Loosely speaking, we our findings are as follows.

6. PLOS authors have the option to publish the peer review history of their article (what does this mean?). If published, this will include your full peer review and any attached files.

Reviewer #1: No

Reviewer #2: No

---

## [Author Response · Author response to Decision Letter 0]

17 Oct 2019

Response to Reviewers

We thank both reviewers for taking the time and effort to read the manuscript and comment on our work. Reviewer 2 considers the manuscript to address a ‘central problem in complex network’ science and recommends publication. Reviewer 1 raises some points regarding our choice of (economic) terminology. We understand the confusion with the terminology in the presented context and have adjusted the terminology, specifically in the title ‘The winner takes it all - Competitiveness of single nodes in globalized supply networks’ and abstract in the direction of network science to more accurately reflect the intent of the manuscript: to quantify the (topological) features that determine the importance of a node in a simplified percolation model inspired by the fundamental forces driving the emergence of global connectivity in a supply network. As such, while we do not aim to exactly model the emergence of globalization, we hope that our results may help to better understand this process and other processes sharing similar mechanisms. We have revised the manuscript to more clearly point out this goal.

We believe the revised manuscript now well matches the level of contribution desired by PLOS ONE.

 

Response to Reviewer 1:

“A serious problem with this paper is that it has very little to do with trade & globalization, and is therefore misleading as far as it's title/advertising goes. The formulation of the problem is a very narrow, cost side approach, whereas "globalization," it's structure and who "wins" (the use of this term is questionable too), involve both demand and supply side considerations. More realistic additions to the model, such as the addition of variable costs on the production side (at the moment there are only fixed costs) and fixed costs on the transportation side (at the moment there are only variable costs) would invalidate the results of the model. In this sense, the model presented in the paper is a caricature of trade and globalization.”

We thank the reviewer for raising this point and apologize for the confusion due to our choice of terminology. We indeed intend the model to be a minimal model (“caricature”) ¬¬of a supply network globalization process capturing only a few key aspects. The model is not intended to exactly represent all aspects of globalization, but focus only on the driving forces in the simplified perspective of a percolation model that remains easy to study even for larger networks. This percolation-type model enables us to study the importance of nodes during the emergence of global connectivity in supply networks from the perspective of network science, a “central problem in complex network” as stated by Reviewer 2. In this context, we understand the reviewer’s confusion with our use of the word ‘globalization’. To avoid confusion and make the aim and scope of the model clearer, we have adjusted the wording in the revised manuscript to use network science terminology rather than economic terms (for example, cluster instead of market). We have also modified the title, now ‘The winner takes it all - Competitiveness of single nodes in globalized supply networks’, and the wording throughout to more clearly convey this intention in the revised manuscript. 

In general, the choice of cost functions and constraints assumed in the model are made for two reasons: (1) to simplify the model as much as possible while keeping the fundamental driving forces; (2) to allow the mapping to a (local) percolation problem with an efficient solution. Nonetheless, many extensions are already included and simply require a rescaling of the parameters or a small (quantitative, not qualitative) modification of the cost functions. In this sense, the results presented describe the qualitative behavior for a range of processes. For a more detailed discussion of the specific points mentioned by the reviewer (additions to the cost function), we refer to the replies below.

We hope that the change in terminology and additional explanations and clarifications more clearly convey the intent and scope of the manuscript.

 “-The supply problem is divorced from the demand side. Why is this about globalization? The authors provide no interpretation or intuition of this.”

As described above, the model is intended to describe the fundamental mechanisms of the growth of a supply network (given fixed demand) from the point of view of a percolation model. We hope that the change in terminology and additional explanations and clarifications more clearly convey the intent and scope of the manuscript.

 “-The production side of the economy is assumed to have a fixed cost, and the economies of scale come from this sole feature. There are no variable costs of production in the model. Such a cost structure is restrictive and applies to very few industries in the real world. There is no explanation for this omission.

-Transportation costs on the other hand are modeled as variable costs. Why? There could very well be fixed costs on the transportation side too. A small change such as this would invalidate the results.”

The reviewer asks about the cost functions in the model and why they were chosen as presented. We thank the reviewer for raising this question. To avoid any potential confusion, we first reiterate the cost structure in the model: 

• transportation costs per unit are linearly increasing proportionally to both distance and amount

• production costs per unit are affine linearly decreasing (economies of scale)

The absolute production cost is then given as a quadratic function of the total amount x as c(x) = c1 x – c2 x2. The production costs describe the effective result of both variable costs increasing linearly with the amount as well as a discounting term due to the economies of scale. In fact, we do not assume explicit fixed costs in production. The specific choice of these cost functions as (affine) linear is made for the simplicity of the model. In particular, other choices for the production costs are possible under reasonable constraints (non-increasing cost functions and identical demand to guarantee the mapping to the percolation problem, see Ref. [29]).

In this cost structure, we implicitly assume that either the fixed costs in production are negligible compared to the variable costs (no fixed costs) or that fixed costs are identical at all nodes. In the second case, the fixed costs only affect the absolute costs, not the ordering of which supplier is cheaper. Similarly, other extensions are already implicitly included in the model. For example, fixed costs in transportation (as suggested by the reviewer) would qualitatively correspond to a non-zero value of the control parameter pT. At this point in the evolution of the model, there may not be a single globally connected cluster. However, the properties identified in the manuscript still help to understand the evolution, the number of large clusters, and to identify nodes that are likely the center of large clusters. As the goal of the manuscript is to study the transition to global connectivity, we scale all transportation costs with the control parameter pT -> 0 to guarantee a single, globally connected cluster in the end.

These additions or modifications would naturally modify the evolution of the model (e.g. slow down or speed up the transition to a single globally connected component). Yet, the fundamental mechanism of larger clusters growing faster until the effects of the economies of scale are balanced by the transport costs at larger distances, remains the same. As such, the results remain qualitatively valid for a broad range of conditions.

We have added a brief discussion of the choice of the cost function, extensions, and the generality of the results in the revised manuscript.

 

“The problem in the paper is more related to a cost-minimization engineering problem, that perhaps fits in logistics or operations research, but is a caricature of the economics of trade and globalization. It is a very narrow and misleading characterization of the problem.”

As discussed above, the manuscript aims to understand the fundamental mechanisms of a supply network globalization process in terms of the importance of the topological features of nodes in the transport network. We concede the point that the presented model is not an exact description of "the economics of trade and globalization". Instead, the model studies the problem in a simplified percolation model including (only) the most relevant driving forces. Transferring the presented results to more complex and realistic economic models, they are, of course, quantitatively varied by the additional influences. However, as explained above for the choice of cost functions, these results are qualitatively relevant to a range of processes. In this sense, we argue that the scope is, in fact, not narrow.

We believe the change in wording and additional explanations in the revised manuscript now more accurately represent the aim and contribution of the work presented in the manuscript.

Response to Reviewer 2:

“In this work, the authors assess the importance of node in trade networks using a percolation-type model of economy of scale. They answer two questions: how a globalized market emerges and who wins the competition and takes the all in the globalization. They conclude two different scenarios depending on the strength of economies of scale. For weak economies of scale, internal properties (costs) of nodes is an important factor while for strong economies of scale, the efficiency is an important factor to find out the power of nodes.

Identification of the power of node in terms of the network structure is a central problem in complex network society and physics of complex systems. In this work, the authors successfully provide a method to identify and analyze the importance of nodes. This work is novel and all the results and discussion in the paper are well supported by their analysis. Therefore, I recommend the paper to be accepted in PLoS ONE with a minor revision taking into account the following issues.”

We thank the reviewer for their favorable judgement and their recommendation to publish the manuscript. We address the individual comments in detail below.

“1. All the analysis in the paper is dependent on a resulting Nash equilibrium state in the model. Is this equilibrium state uniquely defined? If not, is the following analysis reliable?”

We thank the reviewer for this important question. Indeed, multiple Nash equilibria coexist in the process. The simplest example can be seen for sufficiently large economies of scale a and no transportation costs pT = 0: centralized production at any node is a Nash equilibrium.

Importantly, during the evolution of the globalized supply network, the transport costs decrease slowly such that only a single node updates its supplier at first. While this decision may cause a cascade of decisions from other nodes (e.g. when a cluster becomes too small and the cost increase too much to keep all nodes), these decisions are well ordered. For example, an ordering can be defined by the new cost for the node making the decision. Other sensible choices of the ordering (e.g. ordering by the largest cost difference) do not qualitatively change the results. However, these other orderings do not necessarily guarantee that nodes make only ‘local’ changes (i.e., switch to a supplier that supplies one of their neighbors, see ref. [29] for a more detailed discussion).

In this sense (given the ordering of the decisions), while other Nash equilibria exist, the sequence of Nash equilibria that the model goes through is uniquely defined based only on the model parameters and the initial conditions (see ref. [29] for a discussion of the resulting hysteresis effect when reversing the process).

We have added a short explanation on the existence of multiple Nash equilibria to the revised manuscript.

 

“2. The mechanism of merging processes with weak economies of scale is reminiscent the explosive percolation which have been popular in physics community recently. Can the authors add some discussion about a association between them?”

We thank the reviewer for this question. The reviewer correctly identifies the similarities of the network globalization process analyzed in the manuscript to known explosive percolation transitions. The competition between the clusters/markets (effectively not allowing two large clusters to merge until transportation costs decrease sufficiently) is similar to mechanisms in known models of explosive or discontinuous percolation. Additionally, the restriction of transportation costs results in compact clusters, further promoting explosive or discontinuous transitions. In fact, the transition in the presented model is often genuinely discontinuous. The easiest example is the limiting case of very high economies of scale: a single node buying from a new supplier causes all other nodes to also buy from that supplier and the network switches from disconnected nodes to one connected market discontinuously.

A rigorous study of the transition in the presented model in the thermodynamic limit, however, is difficult. In many network topologies, the different scaling of the network size/cluster sizes and the distances in the network breaks the balance between the production cost (economies of scale, cluster sizes) and the transport costs (control parameter, distances), often resulting in trivially discontinuous transitions at pT = 0 in the limit. A sensible study is possible using appropriate network topologies (such as geometrically embedded networks as in the present manuscript or a complete graph) where the corresponding scaling of distances is automatically correct. In these cases, the model in fact exhibits a transition from a slow, gradual transition to a sudden, discontinuous transition.

We have added a short discussion of this connection to the revised manuscript and refer the interested reader (as well as the interested referee) to ref. [29], where the model was first introduced, for a more detailed discussion of the model in a percolation context.

“3. The first sentence in the very last paragraph is not complete: Loosely speaking, we our findings are as follows.”

We thank the reviewer for pointing out the typo and have corrected it.

---

## [Decision Letter · Decision Letter 1]

4 Nov 2019

The winner takes it all - Competitiveness of single nodes in globalized supply networks

PONE-D-19-16528R1

Dear Dr. Han,

We are pleased to inform you that your manuscript has been judged scientifically suitable for publication and will be formally accepted for publication once it complies with all outstanding technical requirements.

With kind regards,

Samir Suweis, Ph.D.

Academic Editor

PLOS ONE

Reviewer's Responses to Questions

**Comments to the Author**

1. If the authors have adequately addressed your comments raised in a previous round of review and you feel that this manuscript is now acceptable for publication, you may indicate that here to bypass the “Comments to the Author” section, enter your conflict of interest statement in the “Confidential to Editor” section, and submit your "Accept" recommendation.

Reviewer #2: All comments have been addressed

2. Is the manuscript technically sound, and do the data support the conclusions?

Reviewer #2: Yes

3. Has the statistical analysis been performed appropriately and rigorously? 

Reviewer #2: Yes

4. Have the authors made all data underlying the findings in their manuscript fully available?

Reviewer #2: Yes

5. Is the manuscript presented in an intelligible fashion and written in standard English?

Reviewer #2: Yes

6. Review Comments to the Author

Reviewer #2: All comments have been successfully addressed. The manuscript has been largely improved so that I recommend the paper to be published in PLOS.

7. PLOS authors have the option to publish the peer review history of their article (what does this mean?). If published, this will include your full peer review and any attached files.

Reviewer #2: No

---

## [Editor Report · Acceptance letter]

11 Nov 2019

PONE-D-19-16528R1 

The winner takes it all - Competitiveness of single nodes in globalized supply networks 

Dear Dr. Han:

I am pleased to inform you that your manuscript has been deemed suitable for publication in PLOS ONE. Congratulations! Your manuscript is now with our production department. 

With kind regards,

on behalf of

Dr. Samir Suweis 

Academic Editor

PLOS ONE